# An Analysis of the Methane Cracking Process for $CO_2$-Free Hydrogen Production Using Thermodynamic Methodologies

**Julles Mitoura dos Santos Junior** [1,*], **Jan Galvão Gomes** [1], **Antônio Carlos Daltro de Freitas** [2] **and Reginaldo Guirardello** [1]

1   School of Chemical Engineering, University of Campinas (UNICAMP), Av. Albert Einstein 500, Campinas 13083-852, SP, Brazil
2   Engineering Department, Exact Sciences and Technology Center, Federal University of Maranhão (UFMA), Av. dos Portugueses, 1966, Bacanga, São Luís 65080-805, MA, Brazil
*   Correspondence: jullesmitoura7@gmail.com; Tel.: +55-(98)-98224-4318

**Abstract:** The thermal cracking process of methane does not present the emissions of polluting gases, forming only hydrogen with a high degree of purity and solid carbon that can be commercialized for other industrial purposes globally. Thermodynamic methodologies based on Gibbs energy minimization and entropy maximization are used in the present study to simulate operating conditions of isothermal and adiabatic reactors, respectively. The chemical equilibrium and combined phases problem were written in a non-linear programming form and optimized with the GAMS software using the CONOPT 3 solver. The results obtained by the methodology described in this study present a good agreement with the data reported in the literature, with mean relative deviations lower than 1.08%. High temperatures and low pressures favor the decomposition of methane and the formation of products. When conditioned in an isothermal reactor, total methane conversions are obtained at temperatures above 1200 K at 1 bar. When conditioned to an adiabatic reactor, due to the lack of energy support provided by the isothermal reactor and taking into account that it is an endothermic process, high methane-conversion rates are obtained for temperatures above 1600 K at 1 bar. As an alternative, the combined effects of the addition of hydrogen to the feed combined with a system of extreme pressure variation indicate a possibility of conducting the thermal cracking process of methane in adiabatic systems. Setting the $CH_4/H_2$ ratio in the system feed at 1:10 at 1600 K and 50 bar, following severe depressurization through an isentropic valve, varying the pressure from 50 to 1 bar, the methane conversion varies from 0 to 94.712%, thus indicating a possible operational conformation for the process so that the amount of carbon generated is not so harmful to the process, taking into account that the formation of the same occurs only after the reaction and heating processes. Under the same operating conditions, it is possible to use about 40.57% of the generated hydrogen to provide energy for the process to occur.

**Keywords:** methane thermal cracking; hydrogen; minimization of Gibbs energy; entropy maximization

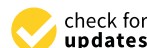



## 1. Introduction

Among the main objectives of modern engineering, the search for sources of energy with low environmental impact has exponentially gained relevance and attention with time. The effects of global warming associated with the means of supplying energy have attracted increasing attention around the world. This problem is directly associated with the world energy matrix, since most of the needed energy sources, at present, are provided by the combustion of fossil sources as shown in Figure 1. Consequently, the excessive use of fossil sources in a conventional way to generate energy implies the release of high volumes of greenhouse gases, and especially carbon monoxide, which presents a constant increase in emission rates year after year as shown in Figure 2.

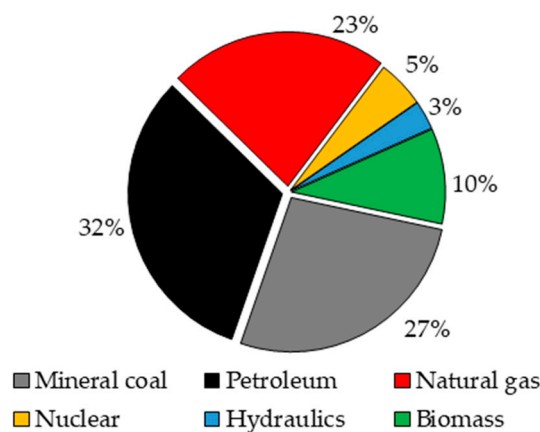

**Figure 1.** World energy matrix with data from IEA, World Energy Outlook 2020 [1].

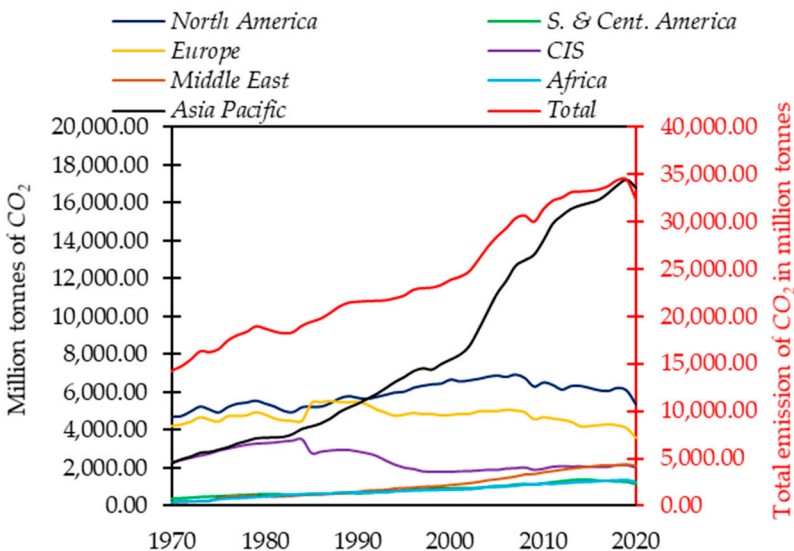

**Figure 2.** $CO_2$ emissions in million tonnes based on data reported by BP, 2020.

There is a reduction in $CO_2$ emission rates between 2019 and 2020, the period of the beginning of the pandemic caused by COVID-19. During the peak of the pandemic, there was a sharp decrease in global $CO_2$ emission rates due to reductions in energy demands and mainly from industrial activities, which decreased by 30% during this period. The effects of the pandemic have led to reductions in $CO_2$ emission rates by more than 5%, being more pronounced in the US (7.6%), India (12.7%), and the UK (19.3%) [2].

The high levels of polluting gases caused by the use of fossil sources has generated considerable interest in the use of alternative energy sources. However, due to the high energy demand, which is predicted to increase by 30% by 2035, and their high availability, the use of fossil fuels will continue to be indispensable for the coming decades. Thus, it is necessary to seek more efficient ways to use fossil fuels, reducing the environmental impacts caused by them.

Several sources of energy free of pollutants are known, such as wind, solar, and nuclear; however, these are inefficient due to high costs and safety issues. Within this scope, hydrogen is considered a clean fuel as it produces only water in its combustion, in addition to its high energy density [3–5]. Hydrogen generates up to three times more energy during combustion (39.4 kWh/kg) compared to any other fuel based on mass (13.1 kWh/kg) [6]. More clearly, a kilogram of hydrogen gas has about the same energy potential as a gallon of gasoline, and this indicates that hydrogen has the potential to replace conventional energy

sources from fossil sources or reduce the dependence of the world energy matrix on these types of energy sources [7].

In 2015, about 48% of the hydrogen generated worldwide was obtained from the natural gas conversion process, a favorable situation considering that this raw material exists in abundance all over the world with about 138,024 thousand million barrels in provenance with data obtained from 2020, as shown in Figure 3 [7].

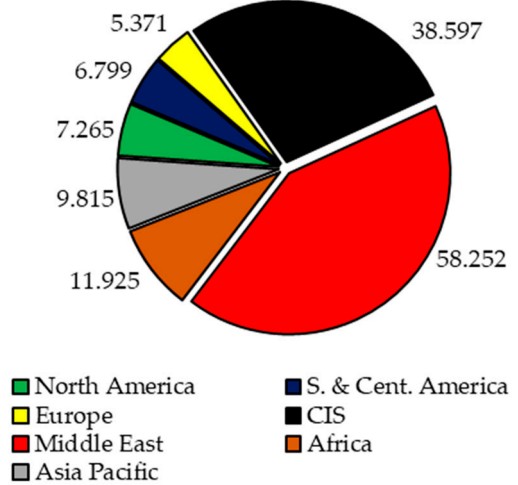

**Figure 3.** Proved natural gas reserves in thousand million barrels based on data reported by BP, 2020.

At present, hydrogen is mainly generated through the steam reforming route of methane, a major component of natural gas (83 to 97%), a process that demands excessive water for its use in the form of steam and presents significant $CO_2$ emissions [8,9]. The socioeconomic situation, at present, and environmental problems that are becoming worse year after year have driven the search for energy generation routes with a low environmental impact, and therefore $CO_2$ emissions during the steam reform process have become one of the main problems of this route. As an alternative to this, the methane cracking process fulfills the objective of being a low-environmental-impact route, since it has zero theoretical emissions of $CO_x$ gases [10]. Figure 4 presents a schematic of the methane cracking process. Since it does not present $CO$ and $CO_2$ formation, it is not necessary to use a *water–gas-shift* reactor and $CO_2$ removal processes.

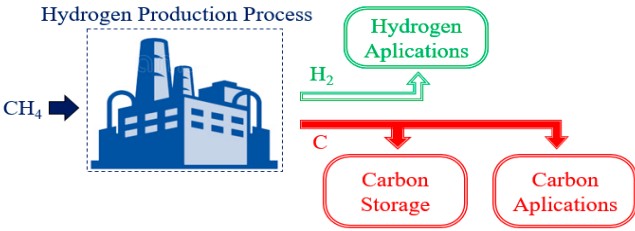

**Figure 4.** Hydrogen production process by methane cracking.

The cracking process is based on the decomposition of the methane molecule into two molecules of hydrogen and one of solid carbon, as presented in Equation (1). It is an endothermic process where the Gibbs energy of the process is zero at 819 K, and to obtain good methane conversion rates, the process must operate at temperatures above 1273 K [11].

$$CH_{4\,(g)} \leftrightarrow 2H_{2(g)} + C_{(s)} \quad \Delta h_\circ^{298.15\,K} = 74.851 \text{kJ/mol} \tag{1}$$

With the objective of producing 100 Mtoe (ton of oil equivalent) of hydrogen, the methane cracking process produces 103.08 Mton of solid carbon, dispensing with the

formation of 255.18 Mton of $CO_2$ that would be generated by the steam reforming process of the methane [11,12].

Table 1 presents the average cost of producing hydrogen using natural gas by the steam reforming and thermal cracking routes. It appears that the most competitive route among those presented is the steam reforming process. This must be implemented together with carbon capture processes to avoid high levels of $CO_2$ emissions and these imply increases of more than 30% in production costs [13,14]. The processes of obtaining hydrogen through the cracking of natural gas were verified by different authors using solar energy as a thermal source. The *SOL-YCARB* project verifies the methane cracking process by direct heating with solar energy. The production costs were estimated to be between 3 to 4 USD/kg of generated $H_2$, which makes this project uncompetitive compared to the methane steam reforming process, which has costs of 2 USD/kg of generated $H_2$ [13,15]. However, the costs calculated for the *SOL-YCARB* project do not include the sale of solid carbon that is generated in large quantities, which should considerably reduce the project costs, since the solid carbon generated during the methane cracking has a wide range of possible applications in rubber products, tires, paints, plastics, and electronics [16].

**Table 1.** Cost of hydrogen production technologies using natural gas.

| Technology | Source | Cost (USD/kg) | Reference |
|---|---|---|---|
| Central steam reforming | Natural gas | 1.5 | [17] |
| Distrib. steam reforming | Natural gas | 2.6 | [17] |
| Pyrolysis/cracking | Natural gas + solar | 3.0 | [15] |
| Pyrolysis/cracking | Natural gas + solar | 3.6 | [18] |
| Pyrolysis/cracking | Natural gas + solar | 4.5 | [15] |
| Steam reforming | Natural gas + solar | 2.2 | [19] |

The fact that it does not produce $CO_x$ gases during the process makes methane cracking a possibility to use fossil sources for the production of energy with a low carbon content. However, the high thermal demands for the reaction to occur is a constraint for its industrial application. In the absence of catalysts, the activation energy of the methane cracking process varies from 356 to 452 kJ/mol [20,21]. This energy demand is considerably reduced for the range of 205 to 236 kJ/mol in the presence of metallic or carbonaceous catalysts, causing the reaction to occur with good methane decomposition rates between 923 to 973 °C using *Mo-Fe/Al$_2$O$_3$* as a catalyst [22,23]. In addition to the high thermal demand, the amount of solid carbon is another barrier to its application, as the solid carbon formed is deposited in the equipment causing clogging in addition to deactivating the catalysts [10]. However, there is a potential market for this by-product, as carbon products derived from methane pyrolysis, such as carbon black, carbon fibers, and carbon nanotubes, have different applications.

In view of the high operational costs of the methane thermal cracking process, it is interesting and necessary to verify the possibility of commercialization of all products generated during this process. Table 2 presents the global market and prices for solid carbon products.

**Table 2.** Global market and estimated prices for solid carbon products [24].

| Carbon Product | Global Markt (tons) | Estimated Price (USD/ton) |
|---|---|---|
| Carbon black | 120,000,000 (2014) | 400–2000 |
| Carbon fibers | 70,000 (2016) | 25,000–113,000 |
| Carbon nanotubes | 5000 (2014) | 100,000–600,000 |

In the absence of catalysts, the carbon product generated during the thermal cracking process is predominantly carbon black. This is important for the metallurgical industry and can be used as a reducing agent and carbon additive in the steel industry. The price of

this depends on its characteristics, for example, carbon black generated from the thermal cracking process of methane for use in tires can be worth around 400 USD/ton, while the market value of special-grade carbon black can exceed 2000 USD/ton. The demand for carbon black was 12 million tonnes in 2014 and is expected to increase to 16.4 million tonnes by 2022 [24,25].

In the absence of catalysts, the carbon product generated during the thermal cracking process is predominantly carbon black. This is important for the metallurgical industry and can be used as a reducing agent and carbon additive in the steel industry. The price of this depends on its characteristics, for example, carbon black generated from the thermal cracking process of methane for use in tires can be worth around 400 USD/ton, while the market value of special-grade carbon black can exceed 2000 USD/ton. The demand for carbon black was 12 million tonnes in 2014 and is expected to increase to 16.4 million tonnes by 2022 [24,25].

Within this context, the present work presented a thermodynamic study of the thermal cracking process of methane in the absence of catalysts using methods based on Gibbs energy minimization and entropy maximization, simulating the operating conditions of isothermal and adiabatic reactors, respectively. In order to provide information for a better elucidation of the process in question, the thermal behavior of the process was verified using the entropy maximization method, since studies using this methodology to verify this process are not reported in the literature, in addition to verifying the possibility to operate it in such a way that the formation of solid carbon during the process, which, although it cannot be avoided, is less harmful to its development.

## 2. Methodology

The resolution methodology used in this text was based on works reported by Dowling and Biegler [26], Marques and Guirardello [27], Freitas and Guirardello [28], and Rossi et al. [29]. It is about analyzing a system from the point of view of chemical equilibrium and combined-phase thermodynamics. With the objective of simulating the operational conditions of isothermal and adiabatic reactors, the problem in question was verified seeking the minimization of the Gibbs energy and the maximization of entropy. These methodologies can be written in the form of nonlinear programming problems defined as an objective function, respecting constraints that assign physical meaning to the problem to evaluate the solutions found and select the optimal solution.

### 2.1. Equilibrium Written as a Nonlinear Gibbs Energy Minimization Problem Simulating an Isothermal Reactor

For reactive systems with components conditioned to constants P and T, the thermodynamic equilibrium condition can be formulated as a Gibbs energy minimization problem (*minG*—Gibbs energy minimization), as shown in Equation (2):

$$\min G = \sum_{i=1}^{NC} n_i^g \mu_i^g + \sum_{i=1}^{NC} n_i^l \mu_i^l + \sum_{i=1}^{NC} n_i^s \mu_i^s \tag{2}$$

The system in the condition of minimum Gibbs energy must obey two restrictions: these are the non-negativity of the number of moles (Equation (3)) and the balance of atoms (Equation (4)).

$$n_i^k \geq 0; i = 1, \dots, NC; k = 1, \dots, NF \tag{3}$$

$$\sum_{i=1}^{NC} a_{mi} \left( \sum_{k=1}^{NF} n_i^k \right) = \sum_{i=1}^{NC} a_{mi} n_i^o, m = 1, \dots, NE \tag{4}$$

The $\mu_i^g$ values of can be calculated from the formation values under reference conditions using the following thermodynamic relationships presented in Equations (5) and (6):

$$\left( \frac{\partial \overline{H}_i^g}{\partial T} \right) = Cp_i^g, i = 1, \ldots, NC \tag{5}$$

$$\frac{\partial}{\partial T} \left( \frac{\mu_i^g}{RT} \right) = -\frac{\overline{H}_i^g}{RT^2}, i = 1, \ldots, NC \tag{6}$$

Considering that the products formed during the methane thermal cracking process are hydrogen and solid carbon, the chemical potentials for the respective phases can be written according to Equations (7) and (8). The methane cracking process occurs at high temperatures and low pressures, which makes it unnecessary to consider the formation of components in the liquid phase. The only component formed in the solid phase during the reaction is coke, so it is reasonable to assume that it behaves ideally.

$$\mu_i^g = \mu_i^0(T, P) + RT \ln P + RT \ln y_i + RT \ln \widehat{\phi}_i \tag{7}$$

$$\mu_i^s = \mu_i^0 \tag{8}$$

To calculate the non-ideality of the gas phase, by determining the fugacity coefficients ($\widehat{\phi}_i$), the virial equation truncated in the second term proposed by Pitzer and Curl [30] modified by Tsonopoulos [31], as presented in Equation (9), is used.

$$\ln \widehat{\phi}_i = \left[ 2 \sum_j^{NC} y_i B_{ij} - B \right] \frac{P}{RT} \tag{9}$$

This methodology has the advantage that the estimation of the second virial coefficient is possible when experimental data are not available. Furthermore, the application of the virial equation presents low mathematical complexity when compared to the cubic state equations, so that the search for the global minimum required in the optimization process can be achieved with less computational effort.

*2.2. Equilibrium Written as a Nonlinear Entropy Maximization Problem Simulating an Adiabatic Reactor*

Under constant P and H conditions, the thermodynamic equilibrium for a reactive multi-component system can be determined by the maximum entropy of the system (*maxS—entropy maximization*), which can be written according to Equation (10) [29]:

$$maxS = \sum_{i=1}^{NC} n_i^g S_i^g + \sum_{i=1}^{NC} n_i^l S_i^l + \sum_{i=1}^{NC} n_i^s S_i^s \tag{10}$$

The entropy maximization methodology must obey the same restrictions used for the Gibbs energy minimization methodology applied to the reactive multicomponent system presented in Equations (3) and (4). However, the maintenance of the system enthalpy is also a restriction (Equation (11)).

$$\sum_{i=1}^{NC} \left( n_i^g \overline{H}_i^g + n_i^l \overline{H}_i^l + n_i^s \overline{H}_i^s \right) = \sum_{i=1}^{NC} n_i^o \overline{H}_i^o = H^o \tag{11}$$

To determine the entropy of each component in the mixture and the enthalpy balance, the thermodynamic relationships presented in Equations (12) and (13) can be used:

$$\overline{S}_i^k = -\left(\frac{\partial \mu_i^k}{\partial T}\right)_{P,n_i^k} \tag{12}$$

$$\frac{\overline{H}_i^k}{RT^2} = -\frac{\partial}{\partial T}\left(\frac{\mu_i^k}{T}\right)_{P,n_i^k}, i = 1,\ldots,NC \tag{13}$$

The formulation of equilibrium as an entropy maximization problem is interesting to determine the equilibrium temperature of the system, mainly in exothermic reactions [32]. The methodology of maximizing the total entropy of the system was used to verify the thermal behavior of the methane thermal cracking reaction simulating the operating conditions of adiabatic reactors.

The methodology of applying the virial equation combined with Gibbs energy minimization and entropy maximization approaches has been widely reported in the literature with very satisfactory results. Freitas and Guirardello [28] presented a thermodynamic approach to biomass gasification reactions using supercritical water as a reaction medium, obtaining excellent results. The reaction system verified by Freitas and Guirardello [28] is of greater complexity when compared to the reactions that occur in the thermal cracking of methane; therefore, this proposal was suitable for the verification of the cracking reaction of methane, combining complexity and robustness to solve the proposed optimization problems.

### 2.3. Mathematical Method for Solving the Models

The calculation of the combined chemical and phase equilibriums described in the previous sections was framed in the convex nonlinear programming model, which guaranteed the existence of a global optimal point [33]. The thermodynamic models discussed in this work were solved in the GAMS 23.9.5 *software*, with the aid of the CONOPT 3 solver, which uses the GRG (*Generalized Reduced Gradient—Abadie, J. and Carpentier, J., 1969*) search method to obtain solutions for nonlinear problems.

The Gibbs energy minimization methodology is widely used to determine the equilibrium compositions of complex reaction systems, obtaining satisfactory results [34–36]. The entropy maximization methodology, although less frequently used, presents excellent results for the verification of equilibrium reaction systems for the determination of equilibrium compositions, in addition to the thermal characterization of the verified systems [29,37,38].

The methane thermal cracking process was verified through the Gibbs energy minimization and entropy maximization models simulating the operating conditions of isothermal and adiabatic reactors, respectively.

### 3. Results and Discussions

#### 3.1. Validation of Methodology

In order to validate the methodologies used in this work, the data for the methane cracking process reported in the literature were used. In the absence of the experimental data that can be used to validate the methodologies, data simulated by Kogan and Kogan [39], data calculated using correlations reported by Ginsburg et al. [40], and data calculated based on kinetic parameters obtained from experiments conducted by Rodat et al. [41] were used.

The data regarding the equilibrium compositions and thermal behavior of the methane cracking process for when it is conditioned to adiabatic reactors are not reported in the literature. However, the research group that developed this article has several publications using Gibbs energy minimization and entropy maximization methodologies validated with experimental data. Freitas and Guirardello [42] discussed the process of oxidative reforming of methane and throughout the text they validated both methodologies with

experimental data in a satisfactory way. That said, the following sections of this text presented evaluations of the methane cracking process using methodologies based on minimization of Gibbs energy and maximization of system entropy, with the objective of presenting the behavior of this process operating in isothermal and adiabatic reactors.

### 3.2. Behavior of the Methane Cracking Process in an Isothermal Reactor (minG)

Figure 5 presents the methane conversion and hydrogen formation as a function of temperature throughout the methane cracking process, calculated using the Gibbs energy minimization methodology. These are verified in the data calculated using correlations to calculate the equilibrium constants reported by Ginsburg et al. [40]. In addition to this, kinetic parameters were presented by Rodat et al. [41] for calculating the methane conversion. Both results show the behavior of the methane cracking process at 1 atm with 100 mol of methane in the feed.

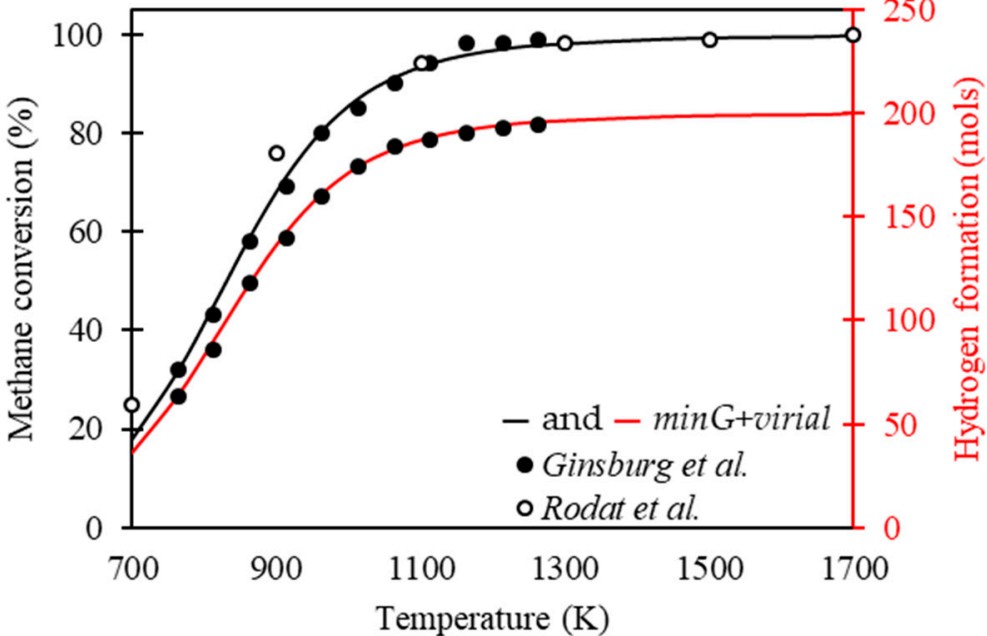

**Figure 5.** Conversion of methane and formation of hydrogen in the equilibrium condition as a function of 1 atm temperature with 100 moles of methane in the feed.

Ginsburg et al. [40] presented correlations for the calculation of equilibrium constants as a function of temperature; thus, the equilibrium compositions of the methane cracking process were calculated. It was verified that the data obtained from the Gibbs energy minimization methodology presented a good fit with the results of Ginsburg et al. [40], with a mean relative deviation of 0.498% for the conversion of methane and 0.552% for the formation of hydrogen.

Rodat et al. [41] presented the development of the kinetic model for the cracking process of methane without solid carbon in the feed for the conditioned system at 1 atm. Both agreed that the reaction can be considered of order 1, and using the kinetic data reported by them it was possible to calculate the compositions for when the system presented behavior close to equilibrium. The data obtained from the Gibbs energy minimization methodology present a good fit with the results of Rodat et al. [41], with a mean relative deviation of 1.072% for methane conversion.

Checking the results reported in Figure 5, the behavior of the methane conversion follows the expected process, presenting conversions close to total for temperatures above 1273 K, following the results presented by Abánades et al. [10]. At this point, there is a possible first negative point of this process: the high thermal demand for the good development of the process.

Figure 6 presents the influence of pressure on the methane cracking process. The results of the mole fraction of methane as a function of temperature are presented for fixed pressure conditions with 1 mol of methane in the feed. The simulated data using the Gibbs energy minimization methodology are presented with the results reported by Kogan and Kogan [39]. These present the mole fraction of methane in the gas phase as a function of temperature for the system operating at 1 atm with 1 mol of methane in the feed. The data were obtained in the thermodynamic equilibrium condition using the *NASA CET-85* program simulating the operational conditions of isothermal reactors.

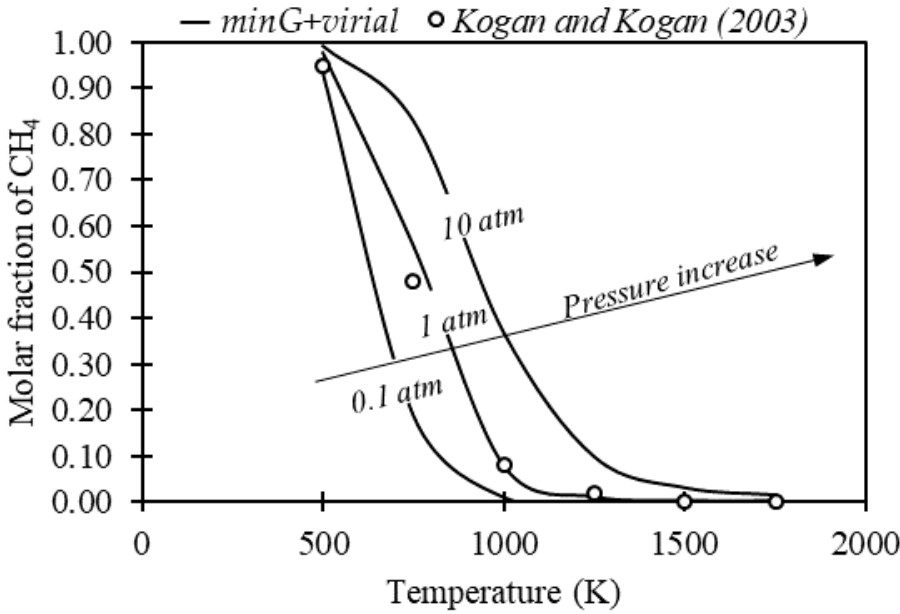

**Figure 6.** Molar fraction of hydrogen as a function of temperature for fixed pressures with 1 mol of methane in the feed [39].

The results presented in Figure 6 indicate a good fit of the data calculated using the Gibbs energy minimization methodology with results reported by Kogan and Kogan [39] with a mean relative deviation of 0.764%. It was verified that the increase in pressure reduced the conversion of methane throughout the process since the molar fraction of methane tended to decrease respecting Le Chatelier's principle, taking into account that the process in question was based on the decomposition of a molecule of methane into two of hydrogen and one of solid carbon, with a higher quantity of molecules in the product compared to the reactants [43–45]. Conditioning the process at 1500 K and 1 mol of methane in the feed and increasing the pressure from 0.1 bar to 10 bar reduces hydrogen formation by 10.931%.

For a better understanding regarding the behavior of the methane thermal cracking process operating in isothermal reactors, it is interesting to verify the influence of the addition of inerts and components that can influence the reaction behavior due to differences in the heat capacities. Figure 7 presents the effect of adding aggregates to the feed along with methane. Figure 7a exhibits the effect of hydrogen addition throughout the methane cracking process. It was verified that the increase in the $CH_4/H_2$ ratio tended to minimize the methane conversion; this result is in agreement with what was observed by Olsik et al. [46] when verifying the effect of hydrogen addition on methane decomposition in isothermal reactors. Considering that it is an isothermal process, the difference in heat capacities between methane and hydrogen justify this behavior.

The inert component considered is nitrogen and its influence on the process is shown in Figure 7b. It is verified that the addition of nitrogen implies higher methane conversion rates throughout the process for the temperature conditions verified in this study. These results

are in agreement with verifications made by Ozalp and Shilapuram Ozalp e Shilapuram [47]. The fact that it behaves as an inert material and maximizes methane conversion during the thermal cracking process makes this a possibility to optimize the process, increasing methane conversion and reducing thermal demand. However, to fully understand the influence of the addition of this component in the process feed, it is also necessary to verify the influence of pressure under a system containing methane and nitrogen in the feed. Considering that lower ratios of $CH_4/N_2$ favor the decomposition of methane, Figure 8 shows the conversion of methane as a function of temperature, conditioning the system at 1 atm with a $CH_4/N_2$ ratio at feed equal to 0.25.

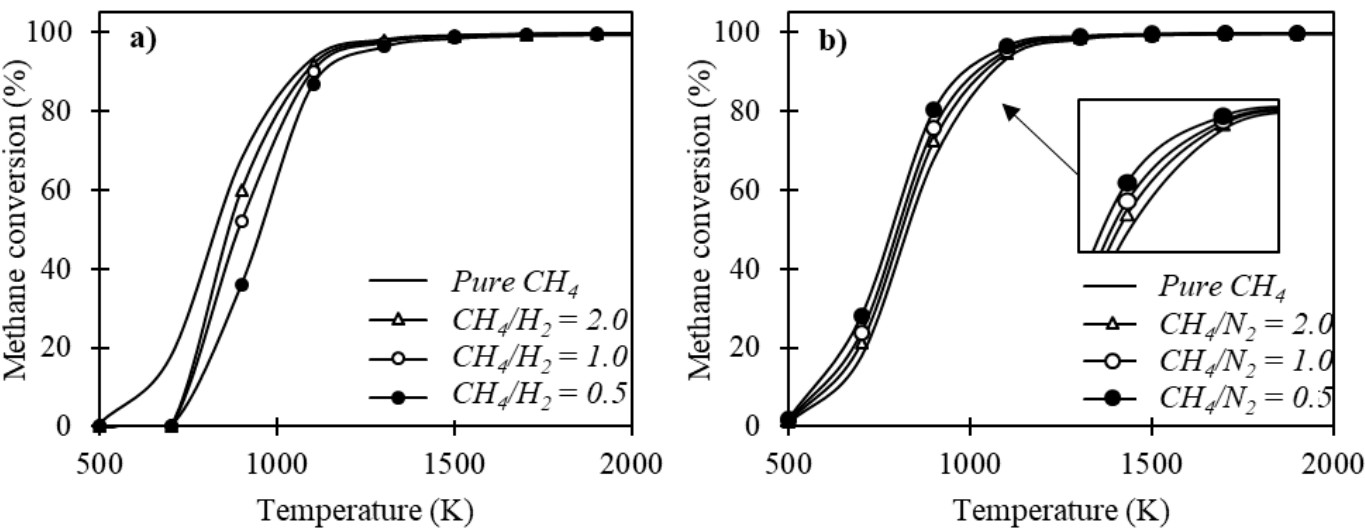

**Figure 7.** Effect of hydrogen (**a**) and nitrogen (**b**) addition on the methane cracking process at 1 atm.

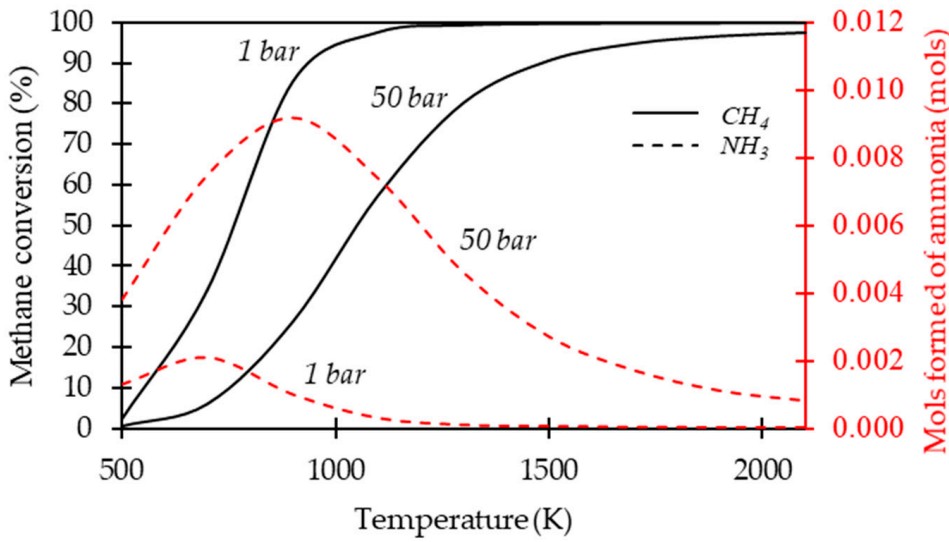

**Figure 8.** Combined effects of nitrogen addition to the feed with pressure on process behavior.

The behavior of the conversion continued to present lower rates with increases in pressure; however, the addition of nitrogen in the feed reduced the thermal demand necessary for methane conversion rates close to totality, requiring 1178 K to convert 99.488% of the methane in the feed. However, it was noted that the pressure increased in favor of the formation of ammonia ($NH_3$) throughout the process. It presented maximum indices for temperatures between 800 and 1000 K with a decreasing profile for temperatures above this. Despite having low ammonia-formation rates, a more elaborate separation process

would be necessary to remove the amounts of unreacted ammonia, nitrogen, and methane from the product stream to obtain pure hydrogen.

The following section presents the behavior of the methane cracking process when conditioned in adiabatic reactors. The entropy maximization methodology simulated the operational conditions of these reactors, providing as a response the equilibrium compositions and the thermal behavior of the process. This discussion is fundamental, since the literature to date concerning the methane cracking process does not present the verification of this process through this methodology, making this work the first to present such results.

### 3.3. Behavior of the Methane Cracking Process in an Adiabatic Reactor (maxS)

Figure 9 presents the thermal behavior of the methane cracking process. For the simulations, the amount of methane was fixed at 1 mol of the process feed. The equilibrium temperature profile followed the expected, since it is an endothermic process; so, the equilibrium temperatures were lower than the initial temperatures for all the conditions verified. It was noted that pressure increases reduced the endothermic behavior of the process; however, increases in the initial temperature minimized the effect of pressure on the thermal behavior of the process. Pressure increases reduced the methane conversion rates; thus, the endothermic effect tended to be reduced and this justifies the results presented in Figure 9.

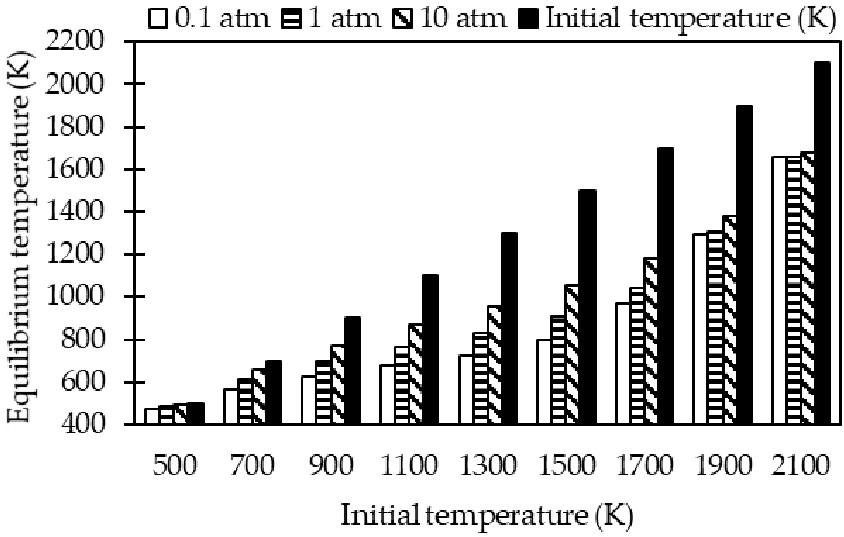

**Figure 9.** Thermal behavior of the methane cracking process as a function of initial temperature and pressure.

As conducted for the verification of the process conditioned to an isothermal reactor, Figure 10 presents the influence of the addition of hydrogen and nitrogen on the thermal behavior of the methane cracking process. The simulations were conducted by conditioning the process to 1 atm.

In general, the addition of hydrogen to the feed reduced the endothermic effect of the process. The differences between the equilibrium temperatures obtained with the addition of hydrogen and nitrogen in the feed with respect to the results obtained using pure methane tended to increase with the increases in temperature. This result is justified by the fact that hydrogen and nitrogen have a lower heat capacity than methane.

Considering that using nitrogen in the feed implies problems with ammonia formation at high pressures, in addition to the needs of more accurate processes for the purification of the product stream, the use of nitrogen as an additional component is no longer interesting. Thus, Figure 11 presents the effect of the addition of hydrogen together with methane in the feed stream on the behavior of the methane cracking process in an adiabatic reactor. The

addition of hydrogen implies lower methane conversion rates for temperatures below 1100 K, a result that was verified for the process operating in an isothermal reactor; however, for temperatures above 1100 K, the process adopted the opposite behavior, presenting higher conversion rates of methane with additions of hydrogen in the feed. Setting the temperature at 1500 K, the addition of 10 mol of hydrogen to the feed with 1 mol of methane increased the methane conversion by 28.347%.

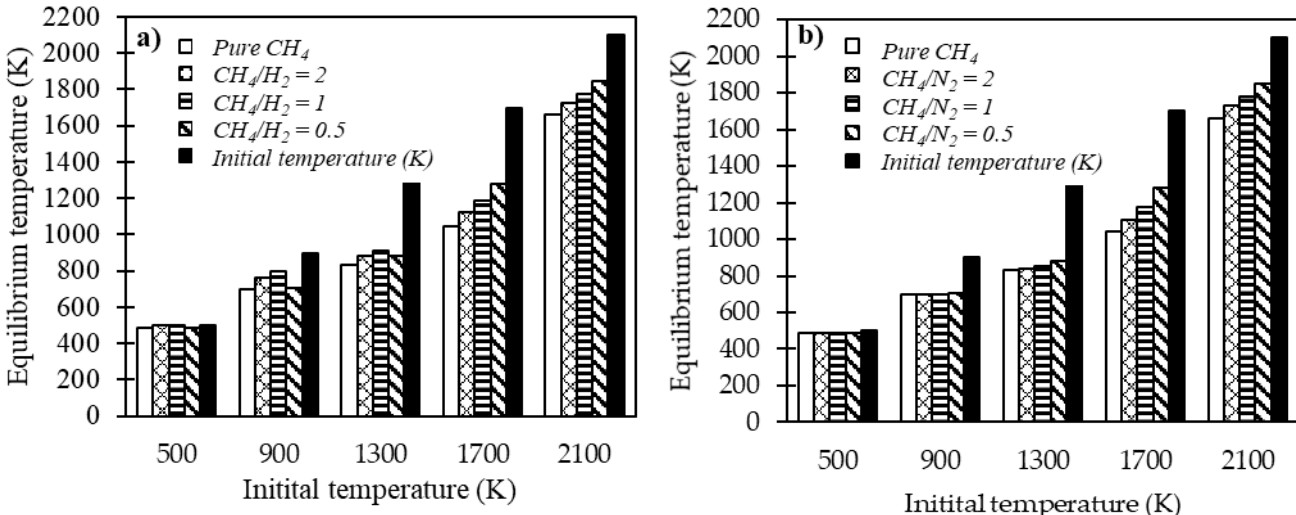

**Figure 10.** Effect of the addition of hydrogen and nitrogen on the thermal behavior of the methane cracking process at 1 atm ((**a**): addition of hydrogen, (**b**): addition of nitrogen).

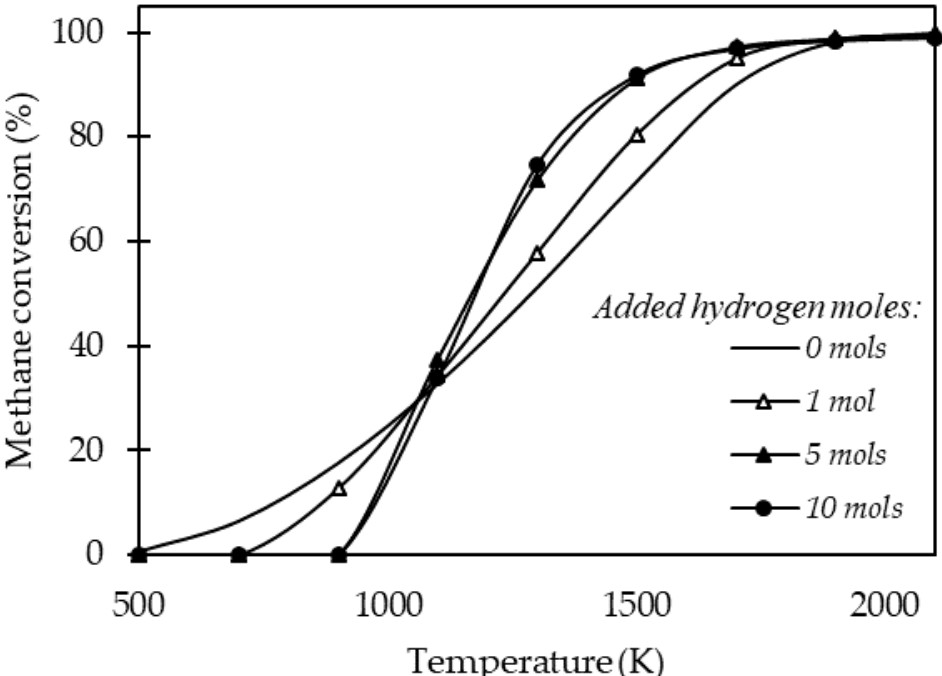

**Figure 11.** Effect of hydrogen addition on the methane cracking process at 1 atm.

The results reported in Figures 10 and 11 indicate that the addition of hydrogen implies gains with respect to the development of the methane cracking process, since it maximizes the methane conversion rates and reduces the endothermic effect of the process. The next step was to check operating conditions with respect to the temperature, pressure, and $CH_4/H_2$ ratio in the process feed.

*3.4. Optimization of the Methane Cracking Process in Adiabatic Reactors*

Figure 12 presents the behavior of the methane conversion as a function of the addition of hydrogen to the feed stream, together with the methane for fixed conditions of temperature and pressure. The simulations were performed using the entropy maximization methodology with 1 mol of methane in the feed. As verified in this text, pressure increases reduced the methane conversion rates, and this result is presented again in Figure 12 where the process is verified operating at 1 and 50 atm under similar operating conditions, showing the higher conversion rates of methane operating at low pressures. Note that when the system is conditioned at 1000 K, the addition of hydrogen in the feed tends to reduce the methane conversion; however, for the results verified at 1500, 1600, and 1700 K, the behavior becomes inverse, where methane conversion increases with the decreasing $CH_4/H_2$ ratio. The objective of this verification was to obtain optimal operating conditions so that it was possible to obtain good methane conversion rates with extreme pressure variations.

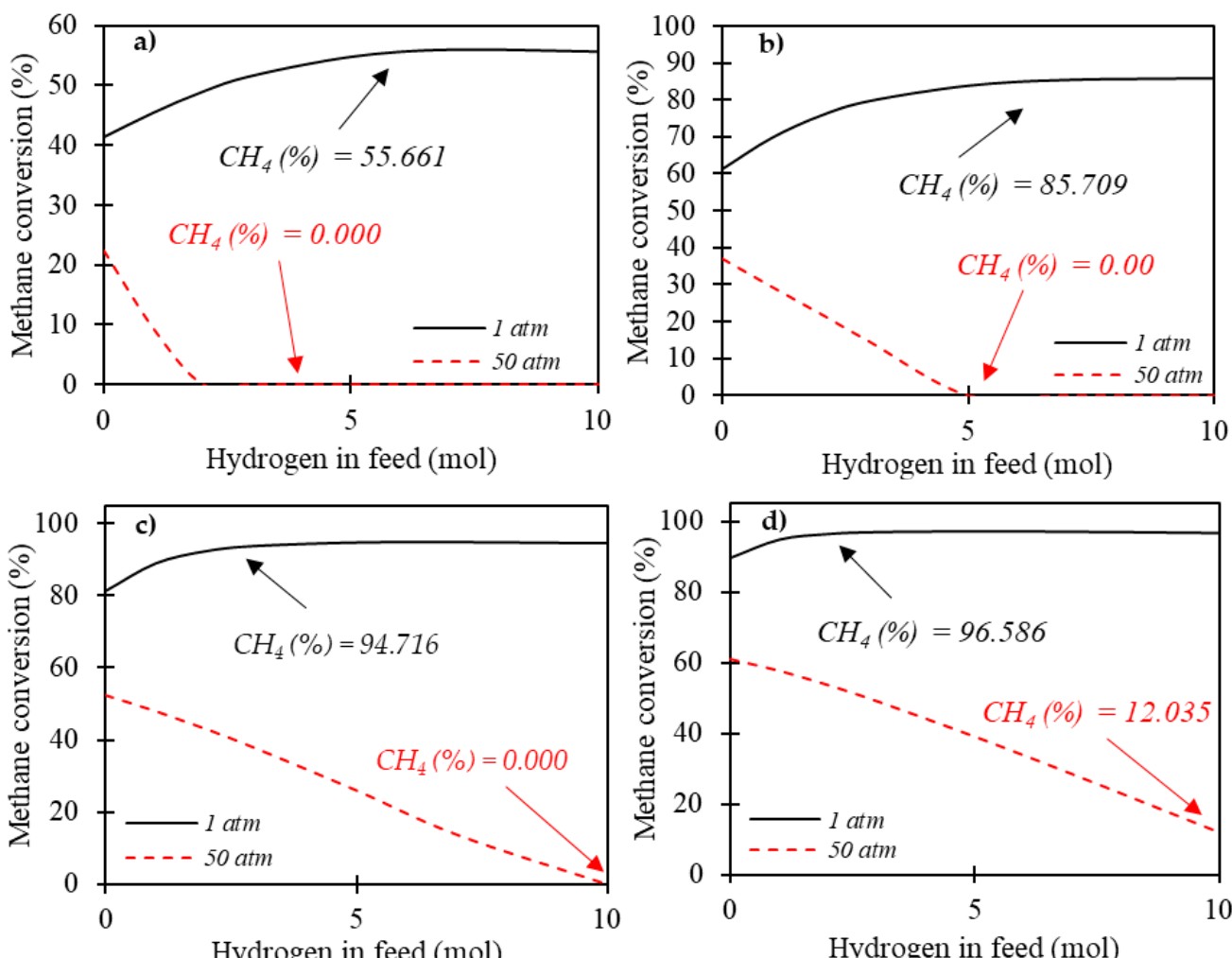

**Figure 12.** Combined effects of hydrogen addition, temperature, and pressure on methane conversions throughout the process ((**a**): 1200 K, (**b**): 1400 K, (**c**): 1600 K, (**d**): 1800 K).

For the verified system with an initial temperature equal to 1200 K, good methane conversion rates were not verified for both the pressure conditions and the amount of hydrogen in the feed verified. Increasing the initial temperature to 1400 K, it was noted that when the system feed was composed of 1 mol of methane with 6 mols of hydrogen, the pressure variation from 50 atm to 1 atm implied a conversion of 85.709 % of the amount

of methane fed into the feed. When the initial temperature of the system was equal to 1600 K, the methane conversion rates were better, being 94.716% for when the system was conditioned at 1 atm and null for the same at 50 atm. Thus, under these operating conditions with a feed composed of 1 mol of methane with 10 mol of hydrogen operating in an adiabatic reactor with an initial temperature of 1600 K, the pressure variation from 50 to 1 atm implied excellent methane conversion rates, when previously, conditioning the system to 50 atm the conversion was null. It was expected that increases in the initial temperature of the problem would present better methane conversion rates and this result is confirmed for when the initial temperature of the process is equal to 1800 K; however, under these conditions, the process presented methane conversion at high pressures, where at 50 atm its conversion was equal to 12.035%, which was no longer interesting for the study in question. Thus, the optimal operating condition of this system was using 1600 K as the initial temperature, a $CH_4/H_2$ ratio in the feed equal to 1:10 with pressure variations of 50 to 1 atm.

Figure 12 presents the results of the theoretical simulation of a methane cracking process based on the extreme pressure variation after heating the stream composed of methane and hydrogen. Figure 13 presents an operational diagram with the optimal result obtained from Figure 12, with a $CH_4/H_2$ ratio in the feed equal to 0.1 initially at 1600 K, varying the pressure from 50 to 1 atm.

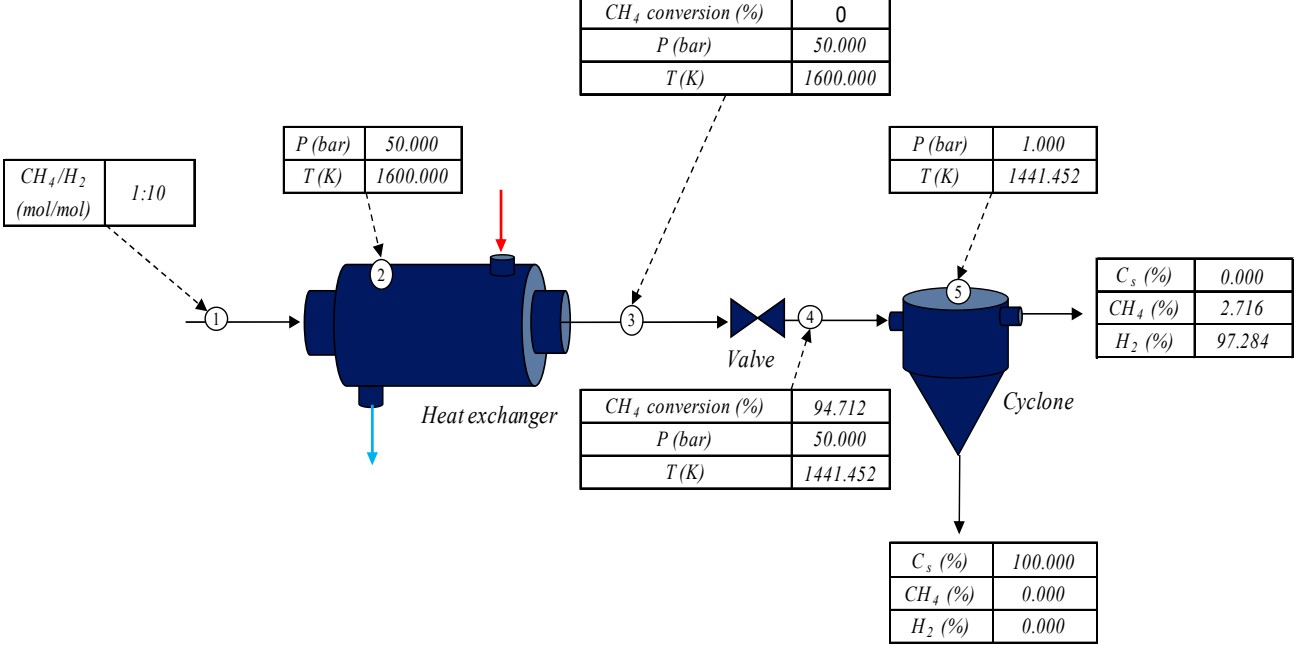

**Figure 13.** Theoretical operational conformation with optimal entropy maximization results.

The feed stream consisting of 1 mol of methane with 10 mol of hydrogen was fed to a heat exchanger that heated the mixture to 1600 K under 50 atm of pressure. Then, the stream heated under high pressure exited the heat exchanger; at this point, the methane conversion was still zero, as can be observed in Figure 12. This traveled to an isentropic depressurization valve, causing a reduction in pressure of 50 to 1 atm, thus causing the conversion of 94.712% of the methane initially introduced into the system. The stream containing the unreacted methane, solid carbon, and generated hydrogen added to the hydrogen inserted at the beginning of the process followed at 1441.452 K to a cyclone that separated the gas stream from the solid carbon generated in the cracking process. The solid carbon exited the cyclone via the bottom stream, and the top stream contained the unreacted methane and hydrogen.

Inserting excess hydrogen into the process feed along with methane added the possibility of operating this process without methane cracking along the heating tube, since methane conversion only occurred after the depressurization of the heated stream. The effect of depressurization can promote the cooling of the methane stream leaving the valve, and for this reason, the addition of hydrogen minimized this effect due to the inverse Joule–Thomson effect, thus reheating the stream leaving the valve, allowing it to be used as a possible thermal utility throughout the process.

The result presented in Figure 13 is expected based on the verifications previously determined in this text, where the effect of the addition of hydrogen to the feed and the pressure on the behavior of the methane conversion were verified. Until the time of publication of this text, studies on the influence of the addition of hydrogen to the feed with methane to verify its influence on the methane cracking process in adiabatic reactors had not been reported in the literature; however, pressure behavior was widely discussed by Ozal et al. [47], Plevan et al. [43], and Younessi-Sinaki et al. [44].

The thermal cracking process of methane presented good rates of hydrogen formation, as verified throughout this text. This product is generated with a high degree of purity and has a high energy density. An alternative for its use is direct combustion, generating only water as a product, as can be observed in Equation (14) [48]:

$$2H_2 + O_2 \rightarrow 2H_2O \quad \Delta h_o^{295.15K} = -286^{kJ}/_{mol} \tag{14}$$

An interesting verification is to try to use part of the generated hydrogen to supply the demands for thermal utilities throughout the process. Taking as reference the operational conformation presented in Figure 13 and the results presented for the behavior of the methane cracking reaction conditioned to an adiabatic reactor, Figure 14 presents the hydrogen formation indices and the amount of hydrogen needed to heat the process through direct combustion and the amount of useful hydrogen.

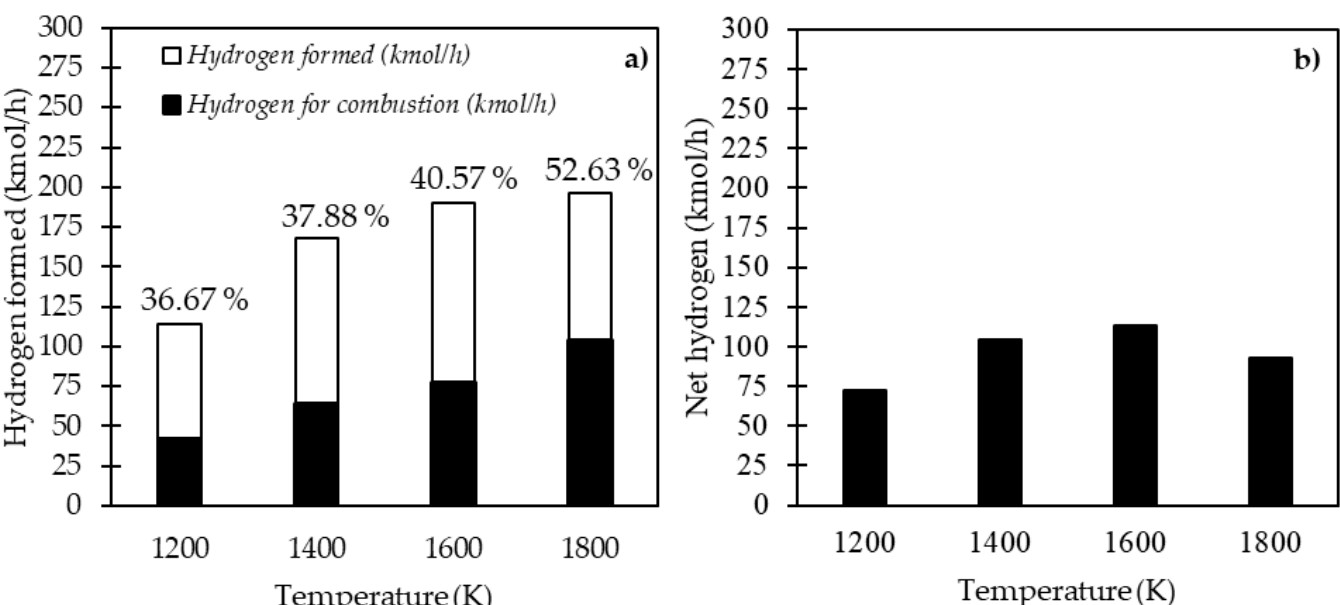

**Figure 14.** Rates of hydrogen formation and consumption for energy generation through hydrogen combustion (**a**) and net hydrogen (**b**).

The results presented were obtained by characterizing the system for processing 100 kmol/h of methane, and the $CH_4/H_2$ ratios in the feed were obtained from the results presented in Figure 12. Conditioning the system at 1200 K, the $CH_4/H_2$ ratio in the feed was 1:6. The temperatures of 1400 K, 1600 K, and 1800 K were equal to 1:6, 1:10, and 1:10, respectively.

As expected, increases in the initial temperature of the system implied a higher demand for the hydrogen generated to be used for heating the system. The optimal operating condition was the one with the highest quantity of liquid hydrogen. For the optimal condition discussed above, at 1600 K, about 52.63% of the hydrogen generated would be destined to supply the thermal utility demands of the process. In this condition, when the system feed stream must be heated up to 1600 K considering that the thermal utilities will be provided with part of the generated hydrogen, the product streams will be 113.31 kmol/h of hydrogen and 84.21 kmol/h of solid carbon.

## 4. Conclusions

Throughout this text, the methane thermal cracking process was verified from a thermodynamic point of view, using methodologies based on the minimization of the Gibbs energy and the maximization of the system entropy in order to seek optimal operating conditions, minimizing the damage caused by the inevitable formation of solid carbon along the process. Thermodynamic modeling was valid with the data reported in the previous literature showing a good approximation with it. The values of the relative mean deviations between the simulated and literature data were, in all verified cases, less than 1.08%.

Increases in temperature combined with reductions in system pressure maximized the hydrogen formation process throughout the methane thermal cracking process. The isothermal reactor had the advantage of constant energy support, because as it is an endothermic process, higher methane conversion rates were obtained, with conversion rates close to 100% for temperatures above 1200 K. When operating in adiabatic reactors, there was the possibility of using hydrogen together with methane in the feed to favor the development of the reaction at temperatures above 1100 K and reduce the endothermic effect of the process. This result poses the possibility for the verification of a possible operational conformation that minimizes possible damages caused by the formation of solid carbon.

The combined effects of adding hydrogen to the system feed with extreme pressure variations allowed for good methane conversion rates, so solid carbon can only be generated after a pressure variation process. Operating with a $CH_4/H_2$ ratio equal to 1:10 in the feed for an initial temperature of 1600 K, it was possible to convert methane from 0 to 94.712% by varying the pressure from 50 to 1 atm, which was the optimal operating condition presented in this text. This possibility indicates a possible operational conformation so that the formation of solid carbon is not so harmful to the process, since the conversion occurs only after the depressurization step, avoiding the formation of solid carbon in the heating system.

Still operating with a $CH_4/H_2$ ratio equal to 1:10 in the feed for an initial temperature of 1600 K, it is possible to use about 40.57% of the hydrogen generated during the process to generate the energy necessary for the same to occur.

**Author Contributions:** J.M.d.S.J., Project proposal. R.G. and A.C.D.d.F., Methodology development. J.M.d.S.J. and J.G.G., Research and Validation. J.M.d.S.J., Development of results. R.G. and A.C.D.d.F., Constant evaluation of results. All authors have read and agreed to the published version of the manuscript.

**Funding:** This research received funding from the National Council for Scientific and Technological Development (CNPq/Brazil—130572/2020-9) and Research Support Foundation of the State of São Paulo (Fapesp/Brazil—2020/03823-1).

**Institutional Review Board Statement:** Not applicable.

**Informed Consent Statement:** Not applicable.

**Acknowledgments:** The authors gratefully acknowledge the financial support received from *CNPq*—Conselho Nacional de Desenvolvimento Científico e Tecnológico (Process 130572/2020-9) and *FAPESP*—Fundação de Amparo à Pesquisa do Estado de São Paulo (Process 2020/03823-1).

**Conflicts of Interest:** The authors declare no conflict of interest.

## Nomenclatures

| | |
|---|---|
| $B$ | Second coefficient of the virial |
| $B_{ij}$ | Second coefficient of the virial for mixture |
| $\hat{\varnothing}_i$ | Fugacity coefficient of component $i$ |
| $R$ | Universal gas constant |
| $G$ | Gibbs energy |
| $H_i{}^k$ | Enthalpy of component $i$ in phase $k$ |
| $H_i{}^0$ | Enthalpy of component $i$ in the standard state |
| $H^0$ | Total enthalpy |
| $S_i{}^k$ | Component $i$ entropy in phase $k$ |
| $S_i{}^0$ | Entropy of component $i$ in the standard state |
| $n_i{}^k$ | Number of moles of component $i$ in phase $k$ |
| $n_i{}^0$ | Number of moles in standard state |
| $a_{mi}$ | Number of atoms of element $i$ in component $m$ |
| $Cp_i$ | Heat capacity of component $i$ |
| $NC$ | Number of components |
| $NF$ | Number of phases |
| $NE$ | Number of elements |
| $\hat{\mu}_i$ | Chemical potential of component $i$ in phase $k$ |
| $y_i$ | Molar fraction of gases |

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
