# Peer review of "An Analysis of the Methane Cracking Process for CO2-Free Hydrogen Production Using Thermodynamic Methodologies"

_methane, doi:10.3390/methane1040020_

Round 1

Reviewer 1 Report

Mitoura et al. provided an interesting study regarding co2 free hydrogen production using thermos methods. Minor revision is suggested before acception, and detailed comments are listed below.

1.     Kindly include more quantitative results in the abstract and conclusion part

2.     Kindly include more updated and related information regarding the cracking and CO2-free application in various discipline, the article below is suggested to be consulted as a starting point. Investigations of CO2 storage capacity and flow behavior in shale formation. Journal of Petroleum Science and Engineering 2022.

3.     Please add more discussion regarding the ratio as described in Fig. 10.

Author Response

Response to Reviewer 1 Comments
Dear reviewer. Thank you very much for your precious contributions.
Point 1:
I have revised the entire text and include quantitative information in the abstract and conclusion.
Abstract:
The thermal cracking process of methane does not present emission of polluting gases, forming only hydrogen with a high degree of purity and solid carbon that can be commercialized for other industrial purposes global. Thermodynamic methodologies based on Gibbs energy minimization and entropy maximization were used to simulate operating conditions of isothermal and adiabatic reactors, respectively. The chemical equilibrium and combined phases problem was written in non-linear programming form and optimized with the software GAMS using the solver CONOPT 3. The results obtained by the methodology described in this work present good agreement with data reported in the literature with mean relative deviations lower than 1.08%. High temperatures and low pressures favor the decomposition of methane and the formation of products. When conditioned in an isothermal reactor, total methane conversions are obtained for temperatures above 1200 K at 1 bar. When conditioned to an adiabatic reactor, due to the lack of energy support provided by the isothermal reactor and taking into account that it is an endothermic process, high methane conversion rates are obtained for temperatures above 1600 K at 1 bar. As an alternative, the combined effects of the addition of hydrogen to the feed combined with a system of extreme pressure variation indicate a possibility of conducting the thermal cracking process of methane in adiabatic systems. Setting the CH4/H2 ratio in the system feed at 1:10 at 1600 K and 50 bar, after severe depressurization through an isentropic valve, varying the pressure from 50 bar to 1 bar, the methane conversion varies from 0 to 94.712%, thus indicating a possible operational conformation for the process so that the amount of carbon generated is not so harmful to the process, taking into account that the formation of the same occurs only after the reaction and heating processes. Under the same operating conditions, it is possible to use about 40.57% of the generated hydrogen to provide energy for the process to occur.
Conclusion:
Throughout this text, the methane thermal cracking process was verified from a thermodynamic point of view, using methodologies based on the minimization of the Gibbs energy and the maximization of the system entropy in order to seek optimal operating conditions, minimizing the damage caused by the inevitable formation of solid carbon along the process. The thermodynamic modeling was valid with data reported in the previous literature showing good approximation with them. The values ​​of the relative mean deviations between the simulated data and the literature data were, in all verified cases, less than 1.08%
Increases in temperature combined with reductions in system pressure maximize hydrogen formation throughout the methane thermal cracking process. The isothermal reactor has the advantage of constant energy support, because as it is an endothermic process, higher methane conversion rates are obtained, with conversion rates close to 100% for temperatures above 1200 K. When operating in reactors adiabatic, there is the possibility of using hydrogen together with methane in the feed to favor the development of the reaction at temperatures above 1100 K and reduce the endothermic effect of the process. This result opens the possibility for the verification of a possible operational conformation that minimizes possible damages caused by the formation of solid carbon.
The combined effects of adding hydrogen to the system feed with extreme pressure variations allow for good methane conversion rates, so solid carbon can only be generated after a pressure variation process. Operating with a CH4/H2 ratio equal to 1:10 in the feed for an initial temperature of 1600 K, it is possible to convert methane from 0 to 94.712% by varying the pressure from 50 to 1 atm, which is the optimal operating condition presented. In this job. This possibility indicates a possible operational conformation so that the formation of solid carbon is not so harmful to the process, since the conversion occurs only after the depressurization step, avoiding the formation of solid carbon in the heating system.
Still operating with a CH4/H2 ratio equal to 1:10 in the feed for an initial temperature of 1600 K, it is possible to use about 40.57% of the hydrogen generated during the process to generate the energy necessary for the same to occur.
Point 2:
I also checked the indication regarding the images. I believe there was a formatting loss but I fixed it for all the images.
Point 3:
Regarding the suggestion to include discussions regarding CO2 storage, I believe that due to the fact that throughout our process we do not have CO2 formation (CH4 -> C + 2H2), discussions about CO2 storage would not fit well in the text.
I hope I have answered all the points and again, thank you very much!

Reviewer 2 Report

The material of the article is important, interesting, especially in the theoretical direction and in the light of modern trends in decarbonization and reducing the impact of greenhouse gases, as well as the synthesis of H2 - a clean fuel that does not emit harmful emissions during use. Practical use, most likely, requires much more detailed research. There are questions regarding the technological problems of coke separation. At what stage is it possible to separate it? Is it necessary to stop the process for this? It is clear that at this stage the theoretical, fundamental part is important. Extremely high process temperatures, work at the simulation stage. All this raises many questions regarding the financial component. What are the general restrictions on the implementation of this process in practice? The unusual design of literary references makes it difficult to get acquainted with literary sources. Figures 2, 5-12, 14 do not indicate dash marks of numerical values on the coordinate axes.

Author Response

Response to Reviewer 2 Comments

Dear reviewer. Thank you very much for your precious contributions.

Point 1:

Dear reviewer. Regarding the coke separation processes, I believe that it is a point to be discussed and also the hydrogen separation process and if it is necessary to use hydrogen or unconverted methane as recycling current and even to use part of the hydrogen generated as fuel. However, the objective of this text was in fact to present a thermodynamic analysis of the methane thermal cracking reaction system. With respect to the characteristics of the equipment involved in the process and to be made of it, this discussion will be reported in a future text.

Regarding the Figures, I believe there was a loss in formatting, which made visualization difficult. These have been fixed.

Reviewer 3 Report

The manuscript entitled "Analysis of methane cracking process for CO2-free hydrogen 2 production using thermodynamic methodologies" reports a numerical study on cracking process of methane subjected to a series of conditions including temperature, pressure and amount of reactants.  Concluding remarks include method validation, effect of temperature, effect of hydrogen addition, optimization of the methane cracking process.  Quality of the manuscript is satisfactory.  Some minor issues should be addressed before acceptance.

1. Ticks should be added to axes of plots in this manuscript.

2. In Eq. (7), (9) and line 239, some abnormal marks are visible, please check if they are correct.

3. "The Figure x" is not proper, e.g., in line 314, it would be better to write "Figure x" directly.

4. What do the arrows mean in Figure 5?

5. In Figure 9 and 10, are black bars redundant? Both the black bars and the horizontal axis represent "Initial temperature".

6. In the calculation, is there a parameter defining the size/volume of the reactor? Because molar amount of reactant is often mentioned, the reviewer think the reaction should occur in a reactor with certain volume. 

Author Response

Response to Reviewer 3 Comments

Dear reviewer. Thank you very much for your precious contributions.

Point 1:

The text lost formatting, so the images were off axis. I fixed the problem by sending the figures in image format.

Point 2:

As for the figures, the equations have lost formatting. I fixed the problem by writing them with the word tool.

Point 3:

Corrections have been made throughout the text as requested.

Point 4:

We agree with the way the figure was presented, which made it difficult to interpret, for this reason, the figure was revised.

Point 5:

At this point, the black bars indicate the initial temperature and these were inserted for the purpose of comparing the equilibrium temperatures with the initial temperatures.

Point 6:

The equation is based on the solution of the combined equilibrium and phase problem in the form of nonlinear programming. The equation does not take into account volume parameters and, therefore, the influence of the reactor volume on the reaction behavior is not discussed. The results presented in this text are the limits for the respective operating conditions.
